# The diagnostic value of pleural fluid homocysteine in malignant pleural effusion

**Jose D. Santotoribio**[1,2]*, **Luis del Valle-Vazquez**[3], **Angela García-de la Torre**[4,5], **Daniel del Castillo-Otero**[6,7], **Juan-Bosco Lopez-Saez**[7,8], **Maria J. Sanchez del Pino**[2]

**1** Department of Laboratory Medicine, Puerto Real University Hospital, Cadiz, Spain, **2** Department of Biomedicine, Biotechnology and Public Health, Cadiz University School of Medicine, Cadiz, Spain, **3** Health Center Las Beatas, Alcala de Guadaira, Sevilla, Spain, **4** Department of Laboratory Medicine, Virgen de la Victoria University Hospital, Malaga, Spain, **5** Instituto de Investigacion Biomédica de Malaga (IBIMA), Malaga, Spain, **6** Department of Pneumology, Puerto Real University Hospital, Cadiz, Spain, **7** Department of Medicine, Cadiz University School of Medicine, Cadiz, Spain, **8** Department of Internal Medicine, Puerto Real University Hospital, Cadiz, Spain

* josediego.santotoribio@uca.es

**Data Availability Statement:** All relevant data are within the paper and its Supporting Information files.

## Abstract

### Background

Pleural fluid homocysteine (HCY) can be useful for diagnosis of malignant pleural effusion (MPE). There are no published studies comparing the diagnostic accuracy of HCY with other tumour markers in pleural fluid for diagnosis of MPE. The aim was to compare the accuracy of HCY with that of carcinoembryonic antigen (CEA), cancer antigen (CA) 15.3, CA19.9 and CA125 in pleural fluid and to develop a probabilistic model using these biomarkers to differentiate benign (BPE) from MPE.

### Methods

Patients with pleural effusion were randomly included. HCY, CEA, CA15.3, CEA19.9 and CA125 were quantified in pleural fluid. Patients were classified into two groups: MPE or BPE. By applying logistic regression analysis, a multivariate probabilistic model was developed using pleural fluid biomarkers. The diagnostic accuracy was determined by receiver operating characteristic (ROC) curves and calculating the area under the curve (AUC).

### Results

Population of study comprised 133 patients (72 males and 61 females) aged between 1 and 96 years (median = 70 years), 81 BPE and 52 MPE. The logistic regression analysis included HCY (p<0.0001) and CEA (p = 0.0022) in the probabilistic model and excluded the other tumour markers. The probabilistic model was: HCY+CEA = Probability(%) = $100 \times (1 + e^{-z})^{-1}$, where $Z = 0.5471 \times [HCY] + 0.3846 \times [CEA] - 8.2671$. The AUCs were 0.606, 0.703, 0.778, 0.800, 0.846 and 0.948 for CA125, CA19.9, CEA, CA15.3, HCY and HCY+CEA, respectively.

**Funding:** The authors received no specific funding for this work.

**Competing interests:** The authors have declared that no competing interests exist.

## Conclusions

Pleural fluid HCY has higher accuracy for diagnosis of MPE than CEA, CA15.3, CA19.9 and CA125. The combination of HCY and CEA concentrations in pleural fluid significantly improves the diagnostic accuracy of the test.

## Introduction

Malignant pleural effusion (MPE) involves the accumulation of exudate in the pleural space due to its invasion by primary tumour cells (mesothelioma) or metastatic tumour cells originating from other tissues. Studies have shown that 42–77% of exudative effusions are secondary to malignancy [1]. The cancers causing MPE, in order of frequency, are lung cancer, breast cancer, lymphoma, unknown primary, genitourinary cancer and gastrointestinal cancer [2]. MPE is found in an estimated 50% of patients with malignant metastatic tumours, either at the time of diagnosis or during the course of their disease [3]. Clinically, the prognosis of MPE is poor, with a median survival of 3 to 12 months, depending on the histological type of the tumour [4,5]. Differentiation between MPE and benign pleural effusion (BPE) is crucial for the treatment and prognosis of these patients. The gold standards for the diagnosis of DPM are pleural fluid cytology and pleura biopsy. The identification of malignant cells in pleural fluid (cytology) is laborious and subjective and shows highly variable sensitivity (from 11% to 78%) [6–8]. Pleural biopsy can achieve high sensitivity, but the procedure is invasive, requires extensive experience, is technically difficult and may not be appropriate in critically ill patients [9]. In contrast, tumour markers (TMs) in pleural fluid may be measured easily and quickly in automated analyzers, being a minimally invasive technique with high accuracy for the diagnosis of DPM.

TMs are molecules produced by normal cells in response to cancer or secreted by the tumour cells themselves that are released into the blood and pleural fluid; their concentrations can be much higher in pleural fluid than in serum [6,10]. Although patients with benign effusions usually have TMs levels in pleural fluid lower than the upper reference limit for serum, parapneumonic and tuberculous pleural exudates can reach serum concentrations up to 14 times the upper reference limit. Moreover, some benign systemic diseases such as renal failure and liver disease can elevate serum TM levels and thereby indirectly elevate TMs concentrations in the pleural fluid as transudative pleural effusions. In addition, some TMs such as cancer antigen (CA) 125 are secreted by normal mesothelial cells and show elevated concentrations in patients with BPE [11]. Thus, the optimal cutoff value of a TM for MPE diagnosis must have sufficiently high specificity to ensure that the marker levels in BPE do not exceed the cutoff point. This high specificity lowers the sensitivity of the TMs. For this reason, some authors recommend the combination of two or more TMs to increase sensitivity without compromising specificity [12,13]. The most frequently used and widely studied pleural fluid TMs are carcinoembryonic antigen (CEA), CA15.3, CA19.9 and CA125. In addition, recent work identified homocysteine (HCY), a sulphur-containing amino acid derived from the metabolism of methionine, as a new TM in pleural fluid, with pleural fluid HCY showing high diagnostic accuracy to distinguish BPE and MPE [14]. However, there are no published studies comparing the diagnostic accuracy of HCY with other TMs in pleural fluid. The aim of this study was to compare the accuracy of HCY with that of CEA, CA15.3, CA19.9 and CA125 in pleural fluid and to develop a probabilistic model using these biomarkers to differentiate MPE from BPE.

## Materials and methods

### Study design and patients

This is a descriptive cross-sectional diagnostic study carried out at Puerto Real University Hospital (Cádiz, Spain) and adhered to the ethical recommendations of the Declaration of Helsinki [15]. The study was approved by the Research Ethics Committee of Cadiz and all participants signed an informed consent form. In cases of patients under the age of 18 years, informed consent form was obtained from the minor's parents or guardians.

White patients treated from January 2014 to January 2017 at Puerto Real University Hospital were studied. This population includes some members of the population assessed in our previous study [14]. The following inclusion criteria were used: patients of any age and sex with pleural effusion and an indication for diagnostic thoracentesis. Exclusion criteria were: patients with previous pleural effusion, pleural fluid extracted from a second or successive thoracocentesis, and purulent pleural fluid.

Included patients were classified into two groups according to the aetiological diagnosis of the pleural effusion: MPE or BPE. The diagnosis of MPE required anatomopathological confirmation by pleural fluid cytology or pleural tissue biopsy.

The objective was to predict a dichotomous qualitative variable (MPE / BPE) using the values of the independent quantitative variables (pleural fluid biomarkers). This objective can be achieved by the multiple logistic regression study. The logistic regression develops a probabilistic model to predict a dichotomous variable using the combination of independent variables [16]. The sample size for the multiple logistic regression study was estimated using the following formula: $n = 10k/p$, where 'k' is the number of independent variables and 'p' is the proportion of patients with MPE [17]. Five pleural fluid biomarkers were included in this study (HCY, CEA, CA15.3, CEA19.9 and CA125). With an expected proportion of patients with MPE of 40%, the estimated sample size was $n = 10 \times 5 / 0.4 = 125$ patients, with a minimum of 75 patients with BPE and 50 patients with MPE. The sampling method was incidental until the estimated sample size was completed.

### Sample analysis

The pleural fluid of each patient extracted by thoracocentesis was analyzed. Biomarkers were quantified in pleural fluid supernatant. To obtain the supernatant, pleural fluid was centrifuged at 4000 revolutions per minute for 5 minutes. HCY was determined by nephelometry in a BNII autoanalyser (Siemens Healthcare Diagnostics, Marburg, Germany), with serum reference values between 4.9 and 15.0 μmol/L. TMs (CEA, CA15.3, CA19.9 and CA125) were quantified by electrochemiluminescence immunoassay in a Hitachi Modular E170 autoanalyser (Roche Diagnostics, Bassel, Switzerland). The serum reference values of TMs are < 5.0 ng/mL for CEA, < 35 U/mL for CA15.3 and CA125, and < 37 U/mL for CA19.9. Although the methods used are not validated for this type of sample, there are many papers using these immunoassays for the determination of TMs in pleural fluid, so these methods are widely accepted. In order to validate the method used for the determination of HCY in pleural fluid, a precision study was performed: repeatability (21 consecutive determinations in the same sample) and reproducibility (one determination per day for 21 days in an aliquot of the same sample conserved at -80˚C).

### Statistical analysis

Data were processed using MedCalc 13.0 (MedCalc Software, Ostend, Belgium), with significance set at $p < 0.05$. The precision study was performed by calculating the coefficient of

variation (CV(%) = 100 x standard deviation / arithmetic mean). Quantitative variables were analysed with the D'Agostino-Pearson test to determine whether they followed a normal (Gaussian) distribution or not. For the descriptive analysis, the frequencies of qualitative variables were used, as well as range and arithmetic mean for normally distributed quantitative variables and range and median for non-Gaussian quantitative variables. Correlations between normally distributed quantitative variables and between non-Gaussian variables were determined using the Pearson correlation coefficient and the Spearman rho correlation coefficient, respectively. Groups were compared using the Student t test for normally distributed variables and using the non-parametric Mann-Whitney U test for non-Gaussian variables. The influence of each independent variable that is, of each of the biomarkers measured in pleural fluid, on the dependent variable (MPE) was evaluated using odds ratios. By applying logistic regression analysis, a multivariate probabilistic model was developed using the independent variables of this study [Probability (%) = $100 \times (1 + e^{-z})^{-1}$, where Z is the constant determined from the pleural fluid biomarkers]. We calculated the diagnostic accuracy of each of the independent variables and of the resulting probabilistic model by analysing receiver operating characteristic (ROC) curves and calculating the area under the curve (AUC), as well as their optimal cutoff value and the corresponding sensitivity and specificity. In order to reduce the number of false positives, the optimal cutoff value was considered to be the one with the highest sensitivity and a specificity > 90%.

## Results

Population of study comprised 133 patients (72 males and 61 females) aged between 1 and 96 years (median = 70 years). In total, 81 patients had BPE and 52 had MPE. All quantitative variables followed a non-gaussian distribution (S1 Table). There were no statistically significant differences between the sexes in terms of the variables studied or any correlation with age (p>0.05). In the precision study, the repeatability and reproducibility CVs were 2.54% and 2.98%, respectively.

Table 1 shows the distribution of patients according to the aetiological diagnosis of the pleural effusion and demographic data; Table 2 shows the descriptive statistics of the biomarkers studied in pleural fluid and the statistical differences between the two patient groups (MPE and BPE); and Table 3 shows the results of the correlation study between the biomarkers quantified in pleural fluid.

The multivariate logistic regression analysis included the pleural fluid concentrations of HCY (p<0.0001) and CEA (p = 0.0022) in the probabilistic model and excluded the other biomarkers. The odds ratios were 1.72 [95% confidence interval (CI) = 1.36–2.18] and 1.46 (95% CI = 1.14–1.87) for HCY and CEA, respectively. The probabilistic model was as follows: Probability (%) = $100 \times (1 + e^{-z})^{-1}$, where Z = 0.5471 × [HCY] + 0.3846 × [CEA]− 8.2671.

The ROC curves of each biomarker and of probabilistic model to differentiate MPE from BPE can be compared in Fig 1. The AUCs and the optimal cutoff values with their corresponding sensitivities and specificities are shown in Table 4. Significant differences were found between the AUC of the probabilistic model and all of the other AUCs of the pleural fluid biomarkers analysed (S2 Table).

## Discussion

The pleural fluid concentrations of HCY, CEA, CA15.3, CA19.9 and CA125 were significantly higher in MPE patients than in BPE patients (Table 2). All pleural fluid biomarkers analysed were directly and proportionally intercorrelated, with the exception of CA125 with CEA and CA19.9. The strongest correlations were found for HCY and CEA with CA15.3 (Table 3).

**Table 1. Distribution of patients according to the aetiological diagnosis of the pleural effusion and demographic data: Median (range) of age and distribution of the sexes.**

| Etiology | n | Age (años) | Sex (n) |
|---|---|---|---|
| Transudative | 42 (31.6%) | 71.0 (40–96) | F: 14 (33.3%); M: 28 (66.7%) |
| Parapneumonic | 24 (18.0%) | 71.5 (18–89) | F: 14 (58.3%); M: 10 (41.7%) |
| Tuberculosis | 4 (3.0%) | 39.0 (31–79) | M: 4 (100%) |
| Thoracic trauma | 4 (3.0%) | 71.5 (65–78) | F: 2 (50.0%); M: 2 (50.0%) |
| Pulmonary embolism | 3 (2.3%) | 40.0 (27–61) | F: 3 (100%) |
| Rheumatoid arthritis | 1 (0.8%) | 53 | M |
| Lupus | 1 (0.8%) | 21 | F |
| Peritonitis | 1 (0.8%) | 52 | M |
| Chylothorax | 1 (0.8%) | 1 | M |
| **Total of BPE** | **81 (60.9%)** | **70.0 (1–96)** | **F: 34 (42.0%); M: 47 (58.0%)** |
| Lung cancer | 22 (16.5%) | 65.0 (52–83) | F: 6 (27.3%); M: 16 (72.7%) |
| Breast cancer | 12 (9.0%) | 67.5 (46–78) | F: 11 (91.7%); M: 1 (8.3%) |
| Lymphoma | 3 (2.3%) | 69.0 (67–72) | F: 1 (33.3%); M: 2 (66.7%) |
| Mesothelioma | 2 (1.5%) | 80.5 (79–82) | F: 1 (50.0%); M: 1 (50.0%) |
| Colon cancer | 2 (1.5%) | 81.0 (79–83) | F: 1 (50.0%); M: 1 (50.0%) |
| Ovarian cancer | 2 (1.5%) | 67.5 (60–75) | F: 2 (100%) |
| Uterus cancer | 2 (1.5%) | 75.0 (69–81) | F: 2 (100%) |
| Melanoma | 2 (1.5%) | 69.0 (66–72) | F: 1 (50.0%); M: 1 (50.0%) |
| Gastric cancer | 1 (0.8%) | 67 | F |
| Multiple myeloma | 1 (0.8%) | 72 | F |
| Unknown primary | 1 (0.8%) | 55 | M |
| Prostate cancer | 1 (0.8%) | 73 | M |
| Thymus cancer | 1 (0.8%) | 65 | M |
| **Total of MPE** | **52 (39.1%)** | **70.0 (46–83)** | **F: 27 (51.9%); M: 25 (48.1%)** |

MPE: malignant pleural effusion; BPE: benign pleural effusion; F: female sex; M: male sex.

In relation to the precision study of the method used for the determination of HCY in pleural fluid, the repeatability and reproducibility CVs were low (<3%), so this method has a very high precision.

The pleural fluid concentration of HCY was the most accurate biomarker for differentiating MPE and BPE (AUC = 0.846), followed by CA15.3 (AUC = 0.800) and CEA (0.778) (Table 4). In the recent study evaluating the pleural fluid concentration of HCY as a TM, which included 89 patients, the AUC obtained was 0.833 [14], which is similar than that obtained in this study. CEA sensitivity was 51.9%, which was somewhat lower than that published in the meta-analysis by Shi et al, which analysed 45 studies and obtained a sensitivity of 54% and specificity of 94% [18]. With CA15.3, a sensitivity of 53.8% was obtained, also slightly lower than that obtained in the meta-analysis by Wu et al [19], which analysed 21 studies and obtained a sensitivity of 58% and specificity of 91%. The optimal cutoff value of the pleural fluid concentration of CEA to differentiate MPE from BPE was similar to the upper reference limit for serum, whereas the cutoff values of HCY, CA15.3 and CA19.9 were lower than their upper reference limits for serum. Pleural fluid CA125 showed a very high optimal cutoff value (1433 U/mL) to differentiate MPE from BPE because of its high concentration in BPE patients, a result of its secretion by normal mesothelial cells [11].

Because the multivariate analysis with five biomarkers only included the pleural fluid concentrations of HCY and CEA in the probabilistic model, these two biomarkers are sufficient to

**Table 2. Descriptive statistics of pleural fluid biomarkers: Median (range); and statistical differences (p value) between patients with MPE and BPE.**

| Etiology | n | HCY μmol/L | CEA ng/mL | CA15.3 U/mL | CA19.9 U/mL | CA125 U/mL |
|---|---|---|---|---|---|---|
| Transudative | 42 | 8.08 (1.82–13.90) | 0.93 (0.20–9.50) | 6.1 (1.0–28.3) | 1.4 (0.6–7.0) | 522 (37.1–2008) |
| Parapneumonic | 24 | 10.18 (3.71–13.60) | 1.43 (0.38–6.90) | 9.9 (1.0–31.0) | 0.85 (0.6–14.6) | 318 (2.8–1098) |
| Tuberculosis | 4 | 10.67 (7.27–13.10) | 0.67 (0.20–1.19) | 13.0 (6.6–30.4) | 2.6 (0.7–27.8) | 340 (78.9–799) |
| Thoracic trauma | 4 | 11.95 (11.40–12.50) | 0.67 (0.28–1.07) | 21.6 (12.8–30.4) | 3.6 (0.6–6.7) | 1504 (629–2380) |
| Pulmonary embolism | 3 | 15.30 (9.80–15.50) | 0.68 (0.60–0.80) | 24.3 (18.3–28.8) | 1.4 (0.6–10.0) | 1346 (853–1390) |
| Rheumatoid arthritis | 1 | 9.62 | 2.65 | 11.3 | 1.9 | 92.8 |
| Lupus | 1 | 7.00 | 0.20 | 8.3 | 4.8 | 696 |
| Peritonitis | 1 | 8.20 | 0.44 | 6.9 | 0.6 | 436 |
| Chylothorax | 1 | 8.03 | 0.70 | 3.4 | 3.8 | 390 |
| **Total of BPE** | **81** | **8.39 (1.82–15.50)** | **0.93 (0.20–9.50)** | **8.2 (1.0–31.0)** | **1.4 (0.6–27.8)** | **508 (2.8–2380)** |
| Lung cancer | 22 | 12.90 (9.89–16.50) | 10.10 (0.66–184) | 31.2 (7.4–208) | 7.6 (0.6–1562) | 541 (32.4–7305) |
| Breast cancer | 12 | 10.24 (9.28–25.20) | 29.65 (0.51–210) | 105 (9.6–300) | 15.5 (0.6–306) | 709 (97.1–1884) |
| Lymphoma | 3 | 16.50 (15.00–23.30) | 0.44 (0.20–2.50) | 9.8 (5.7–18.3) | 0.6 (0.6–1.2) | 1603 (761–2016) |
| Mesothelioma | 2 | 18.30 (12.20–24.40) | 0.40 (0.20–0.60) | 28.5 (21.6–35.5) | 3.0 (2.5–3.6) | 132 (82.0–182) |
| Colon cancer | 2 | 27.30 (20.30–34.30) | 5.26 (0.42–10.10) | 5.9 (5.2–6.6) | 7.2 (2.7–11.8) | 860 (666–1055) |
| Ovarian cancer | 2 | 13.25 (10.00–16.50) | 44.48 (0.50–88.47) | 86.9 (35.6–138) | 278.3 (3.9–552.7) | 6987 (2375–11599) |
| Uterus cancer | 2 | 20.55 (16.70–24.40) | 38.43 (4.87–72.00) | 27.1 (5.8–48.5) | 279 (268–290) | 262 (116–407) |
| Melanoma | 2 | 22.40 (20.70–24.10) | 0.40 (0.20–0.60) | 8.1 (4.6–11.6) | 1.0 (0.7–1.4) | 419 (350–489) |
| Gastric cancer | 1 | 25.50 | 4207 | 50.3 | 4059 | 9677 |
| Multiple myeloma | 1 | 15.40 | 0.70 | 52.3 | 2.9 | 98.0 |
| Unknown primary | 1 | 10.30 | 979 | 300 | 559 | 2116 |
| Prostate cancer | 1 | 9.36 | 0.65 | 13.2 | 2.9 | 594 |
| Thymus cancer | 1 | 16.80 | 1.60 | 42.7 | 4.4 | 1526 |
| **Total of MPE** | **52** | **13.75 (9.28–34.30)** | **8.80 (0.20–4207)** | **33.3 (4.6–300)** | **6.0 (0.6–4059)** | **630 (32.4–11599)** |
| p value* | 133 | <0.0001 | <0.0001 | <0.0001 | <0.0001 | 0.0377 |

MPE: malignant pleural effusion; BPE: benign pleural effusion; CEA: carcinoembryonic antigen; CA: cancer antigen; HCY: homocysteine

*U Mann-Whitney test.

obtain the maximum diagnostic accuracy for MPE. The odds ratio of HCY was higher than that of CEA. Although CA15.3 had a high diagnostic accuracy (AUC = 0.800), its exclusion from the probabilistic model may be due to its strong correlation with HCY and CEA (Table 3). The diagnostic accuracy obtained with the probabilistic model (AUC = 0.948) was significantly higher than that obtained with each individual biomarker (S2 Table). The combination of HCY and CEA in pleural fluid improved the diagnostic accuracy of the test. The probabilistic model (HCY+CEA) achieved high sensitivity (86.5%) and specificity (97.5%) for

**Table 3. Spearman's coefficient of rank correlation (rho) between the biomarkers quantified in pleural fluid (n = 133).**

|  | CEA | CA15.3 | CA19.9 | CA125 |
|---|---|---|---|---|
| **HCY** | 0.174 (p = 0.0459) | 0.386 (p<0.0001) | 0.245 (p = 0.0045) | 0.193 (p = 0.0270) |
| **CEA** | - | 0.416 (p<0.0001) | 0.365 (p<0.0001) | p>0.05 |
| **CA15.3** | - | - | 0.273 (p = 0.0017) | 0.270 (p = 0.0019) |
| **CA19.9** | - | - | - | p>0.05 |

CEA: carcinoembryonic antigen; CA: cancer antigen; HCY: homocysteine.

differentiating MPE and BPE (Table 4). The study by Porcel et al, which analysed 416 patients with pleural effusion, obtained a sensitivity of 54% through the combination of four TMs (CA15.3, CA19.9, CA125 and CYFRA 21–1) [20]; the work by Gaspar et al, which included 155 patients, achieved a sensitivity of 75% through the combination of three TMs (CEA, CA15.3 and TAG-72) [21]; and the meta-analysis by Yang et al, which analysed combinations of pleural fluid TMs for the diagnosis of MPE in 20 studies, obtained a sensitivity of 64% through the combination of CEA+CA15.3 and of 58% with CEA+CA19.9 [22]. All of these

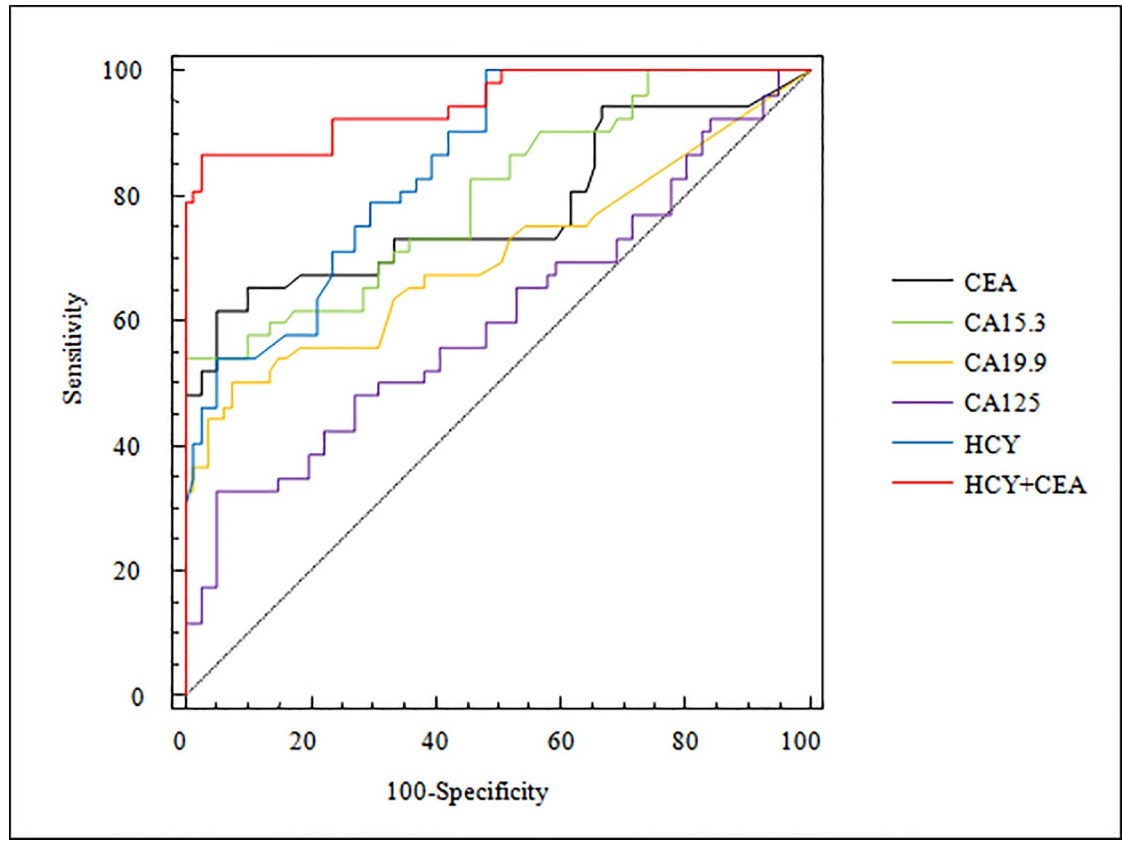

**Fig 1. The ROC curves of pleural fluid biomarkers and probabilistic model for the diagnosis of malignant pleural effusion (n = 133).** CEA: carcinoembryonic antigen; CA: cancer antigen; HCY: homocysteine; HCY+CEA: probabilistic model (%) = 100 x $(1 + e^{-z})^{-1}$, Z = 0.5471 x [HCY] + 0.3846 X [CEA]– 8.2671.

**Table 4. AUC, optimal cutoff values, sensitivity and specificity of HCY, CEA, CA15.3, CA19.9, CA125, and probabilistic model (HCY+CEA) for the diagnosis of malignant pleural effusion (n = 133).**

|  | AUC (95% CI) | Cutoff | Sensitivity (95% CI) | Specificity (95% CI) |
|---|---|---|---|---|
| CA125 | 0.606 (0.517–0.689) | 1433 U/mL | 32.7% (20.3–47.1) | 95.1% (87.8–98.6) |
| CA19.9 | 0.703 (0.617–0.779) | 7.0 U/mL | 50.0% (35.8–64.2) | 92.6% (84.6–97.2) |
| CEA | 0.778 (0.698–0.846) | 5.88 ng/mL | 51.9% (37.6–66.0) | 95.1% (87.8–98.6) |
| CA15.3 | 0.800 (0.722–0.864) | 24.7 U/mL | 53.8% (39.5–67.8) | 90.1% (81.5–95.6) |
| HCY | 0.846 (0.773–0.902) | 13.60 μmol/L | 53.8% (39.5–67.8) | 95.1% (87.8–98.6) |
| HCY+CEA | 0.948 (0.896–0.979) | 46.54% | 86.5% (74.2–94.4) | 97.5% (91.3–99.6) |

AUC: area under the curve; CI: confidence interval; CEA: carcinoembryonic antigen; CA: cancer antigen; HCY: homocysteine; HCY+CEA: probabilistic model (%) = $100 \times (1 + e^{-z})^{-1}$; $Z = 0.5471 \times [HCY] + 0.3846 \times [CEA] - 8.2671$.

studies achieved a lower sensitivity than we obtained here using the combination of HCY +CEA.

These results show an increase of pleural fluid HCY concentration in patients with MPE. The metabolism of HCY occurs via two pathways, via remethylation to methionine using 5-methyltetrahydrofolate (vitamin B9 or folate) as methyl group donor and cobalamin (vitamin B12) as cofactor or via transsulphuration using pyridoxine (vitamin B6) as cofactor, which leads to the degradation of HCY to cysteine [23]. The deficiency of these vitamins B in serum is associated with hyperhomocysteinemia [24]. The increased of pleural fluid HCY concentration in patients with MPE may be due to elevated cellular consumption of vitamins B6, B9 and B12 caused by the intense metabolism of the tumour cells present in the pleural space. The lack of vitamins B can block the two metabolic pathways of HCY and cause its accumulation in the pleural fluid. Moreover, in some studies, serums TMs were superior to pleural fluid TMs for diagnosis of MPE [25]. Further studies are needed to research the association between vitamins B and HCY in pleural fluid, and to assess the diagnostic value of serum HCY and its pleural fluid/serum ratio in patients with MPE.

## Conclusions

Pleural fluid HCY can be considered a TM with high power to differentiate MPE from BPE. Pleural fluid HCY concentration may be measured easily and quickly in automated analyzers and could be a biomarker commonly used for diagnosis of MPE. Pleural fluid HCY has higher accuracy than CEA, CA15.3, CA19.9 and CA125 for diagnosis of MPE. The combination of HCY and CEA in pleural fluid significantly improves the diagnostic accuracy of the test. New studies are needed to confirm these conclusions.

## Supporting information

**S1 Table. Descriptive statistics of independent variables (n = 133).**
(DOC)

**S2 Table. Comparison of ROC curves.** Difference between AUCs.
(DOC)

## Author Contributions

**Conceptualization:** Jose D. Santotoribio.

**Data curation:** Jose D. Santotoribio, Luis del Valle-Vazquez, Angela García-de la Torre, Daniel del Castillo-Otero, Juan-Bosco Lopez-Saez.

**Formal analysis:** Jose D. Santotoribio, Luis del Valle-Vazquez.

**Investigation:** Jose D. Santotoribio, Luis del Valle-Vazquez, Angela García-de la Torre, Daniel del Castillo-Otero, Juan-Bosco Lopez-Saez, Maria J. Sanchez del Pino.

**Writing – original draft:** Jose D. Santotoribio.

**Writing – review & editing:** Jose D. Santotoribio, Luis del Valle-Vazquez, Angela García-de la Torre, Daniel del Castillo-Otero, Juan-Bosco Lopez-Saez, Maria J. Sanchez del Pino.

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
