## [Decision Letter · Decision Letter 0]

27 Jul 2019

PONE-D-19-17055

The diagnostic value of pleural fluid homocysteine in malignant pleural effusion

PLOS ONE

Dear Dr. Santotoribio,

Thank you for submitting your manuscript to PLOS ONE. After careful consideration, we feel that it has merit but does not fully meet PLOS ONE’s publication criteria as it currently stands. Therefore, we invite you to submit a revised version of the manuscript that addresses the points raised during the review process.

We would appreciate receiving your revised manuscript by Sep 10 2019 11:59PM. To enhance the reproducibility of your results, we recommend that if applicable you deposit your laboratory protocols in protocols.io, where a protocol can be assigned its own identifier (DOI) such that it can be cited independently in the future. For instructions see: http://journals.plos.org/plosone/s/submission-guidelines#loc-laboratory-protocols

We look forward to receiving your revised manuscript.

Kind regards,

Luka Brcic

Academic Editor

PLOS ONE

Journal Requirements:

1) As your study includes data from minors under the age of 18, please provide additional details regarding participant consent. In the ethics statement in the Methods and online submission information, please ensure that you have specified whether you obtained consent from the minors' parents or guardians.

2) Please note that according to our submission guidelines (http://journals.plos.org/plosone/s/submission-guidelines), outmoded terms and potentially stigmatizing labels should be changed to more current, acceptable terminology. For example: “Caucasian” should be changed to “white” or “of [Western] European descent” (as appropriate).

3) In your Methods section, please provide additional information about the participant recruitment method and the demographic details of your participants. Please ensure you have provided sufficient details to replicate the analyses such as: a) a table of relevant demographic details, b) a statement as to whether your sample can be considered representative of a larger population, c) a description of how participants were recruited, and d) descriptions of where participants were recruited and where the research took place.

4) We recommend that figure 1 is provided in colour, rather than black and white, so that the individual ROC curves are more easily distinguished.

Reviewers' comments:

Reviewer's Responses to Questions

**Comments to the Author**

1. Is the manuscript technically sound, and do the data support the conclusions?

Reviewer #1: Yes

Reviewer #2: Yes

2. Has the statistical analysis been performed appropriately and rigorously? 

Reviewer #1: Yes

Reviewer #2: Yes

3. Have the authors made all data underlying the findings in their manuscript fully available?

Reviewer #1: Yes

Reviewer #2: No

4. Is the manuscript presented in an intelligible fashion and written in standard English?

Reviewer #1: Yes

Reviewer #2: Yes

5. Review Comments to the Author

Reviewer #1: The manuscript is scientific research with data that supports the conclusions. Statistical analysis has been performed rigorously. The manuscript is written in correct English. Conclusion follows results.

Reviewer #2: Thank you for the opportunity to review this paper regarding the use of pleural fluid tumour markers and homocysteine to identify malignancy as a cause of the effusion.

I find the paper generally very sound and the findings of potential clinical importance. I have a number of comments and suggestions.

Major comments.

I think a sentence at a reference (at least one of each) is needed to describe the probabilistic model. This could be included in the introduction or with a relevant explanatory reference in the section on statistical analysis. This is important for others who may wish to confirm this work as well as understanding comparisons with other papers where different statistical processes may have been used.

The authors need to specify that the assays were performed in a sample type for which the assays are not validated. This raises the issue of their use being “off label” described in different parts of the world as “in house in-vitro diagnostic devices” or “lab derived tests”. At the very least this needs to be acknowledged. There should also be a cautionary note that results may be different in different assays and that formal validation of technical aspects may be needed. Specifically Boot et al (Clinical Chemistry 2010;56:8 1351–1361) have shown non-linear dilution of CEA and CA 19.9 in pancreatic cyst fluid samples using the Roche assay suggesting that measurements in neat samples (other than serum) may suffer some interference. If the authors have validated the analytical performance of these assays in these fluids that would be a benefit.

A brief consideration of the nature of the reference standard (cytology or biopsy) is suggested. This may include a consideration of possible weaknesses in these reference standards as well as a comment on why the use of tumour markers is advantageous over these standards.

Minor comments

Page 4, line 95. I suggest putting a reference for the 1993 declaration of Helsinki rather than the local and date in parentheses.

Page 6, line 151. If this population includes the members of the population assessed in reference 14 this should be stated so that the data sets are not considered to be separate for meta-analysis or review.

Page 8 table 2. There appears to be a line missing for the correlation between CA 19.9 and CA 125.

Page 8, data in line 172 and following. The number of significant figures seems excessive. 2 decimal places should be sufficient for the odds ratios and their CI.

Page 8, line 171. Should a cutoff be supplied for the odds ratios for HCY and CEA?

Page 9, line 199. I suspect that he AUC from this paper and the previous paper are not significantly different. I would describe them as similar rather than the previous one being slightly lower (perhaps “marginally lower” may be more accurate.

Page 9. Line 214. The AUC is missing a leading “0.”.

6. PLOS authors have the option to publish the peer review history of their article (what does this mean?). If published, this will include your full peer review and any attached files.

Reviewer #1: Yes: Silvana Smojver-Ježek

Reviewer #2: Yes: Graham RD Jones

---

## [Author Response · Author response to Decision Letter 0]

30 Aug 2019

Response to Reviewers

Journal Requirements

1) As your study includes data from minors under the age of 18, please provide additional details regarding participant consent. In the ethics statement in the Methods and online submission information, please ensure that you have specified whether you obtained consent from the minors' parents or guardians.

Changes in the manuscript:

Materials and Methods, Study design and patients: “The study was approved by the Research Ethics Committee of Cadiz and all participants signed an informed consent form. In cases of patients under the age of 18 years, informed consent form was obtained from the minor’s parents or guardians.”

2) Please note that according to our submission guidelines (http://journals.plos.org/plosone/s/submission-guidelines), outmoded terms and potentially stigmatizing labels should be changed to more current, acceptable terminology. For example: “Caucasian” should be changed to “white” or “of [Western] European descent” (as appropriate).

Changes in the manuscript:

Materials and Methods, Study design and patients: “White patients treated from January 2014 to January 2017 at Puerto Real University Hospital were studied.”

3) In your Methods section, please provide additional information about the participant recruitment method and the demographic details of your participants. Please ensure you have provided sufficient details to replicate the analyses such as: a) a table of relevant demographic details, b) a statement as to whether your sample can be considered representative of a larger population, c) a description of how participants were recruited, and d) descriptions of where participants were recruited and where the research took place.

Changes in the manuscript:

Materials and Methods, Study design and patients: “This is a descriptive cross-sectional diagnostic study carried out at Puerto Real University Hospital (Cádiz, Spain)… The sampling method was incidental until the estimated sample size was completed.”

Results: “Table 1 shows the distribution of patients according to the aetiological diagnosis of the pleural effusion and demographic data” (A table with the demographic data of the patients studied has been added).

4) We recommend that figure 1 is provided in colour, rather than black and white, so that the individual ROC curves are more easily distinguished.

Changes in the manuscript:

Figure 1 in colour

Review Comments to the Author

Reviewer #1: The manuscript is scientific research with data that supports the conclusions. Statistical analysis has been performed rigorously. The manuscript is written in correct English. Conclusion follows results.

Reviewer #2: Thank you for the opportunity to review this paper regarding the use of pleural fluid tumour markers and homocysteine to identify malignancy as a cause of the effusion. I find the paper generally very sound and the findings of potential clinical importance. I have a number of comments and suggestions.

Major comments.

1) I think a sentence at a reference (at least one of each) is needed to describe the probabilistic model. This could be included in the introduction or with a relevant explanatory reference in the section on statistical analysis. This is important for others who may wish to confirm this work as well as understanding comparisons with other papers where different statistical processes may have been used.

Changes in the manuscript:

Materials and Methods, Study design and patients: “The objective was to predict a dichotomous qualitative variable (MPE / BPE) using the values of the independent quantitative variables (pleural fluid biomarkers). This objective can be achieved by the multiple logistic regression study. The logistic regression develops a probabilistic model to predict a dichotomous variable using the combination of independent variables [16].”

2) The authors need to specify that the assays were performed in a sample type for which the assays are not validated. This raises the issue of their use being “off label” described in different parts of the world as “in house in-vitro diagnostic devices” or “lab derived tests”. At the very least this needs to be acknowledged. There should also be a cautionary note that results may be different in different assays and that formal validation of technical aspects may be needed. Specifically Boot et al (Clinical Chemistry 2010;56:8 1351–1361) have shown non-linear dilution of CEA and CA 19.9 in pancreatic cyst fluid samples using the Roche assay suggesting that measurements in neat samples (other than serum) may suffer some interference. If the authors have validated the analytical performance of these assays in these fluids that would be a benefit.

Changes in the manuscript:

Materials and Methods, Sample analysis: “Although the methods used are not validated for this type of sample, there are many papers using these immunoassays for the determination of TMs in pleural fluid, so these methods are widely accepted. In order to validate the method used for the determination of HCY in pleural fluid, a precision study was performed: repeatability (21 consecutive determinations in the same sample) and reproducibility (one determination per day for 21 days in an aliquot of the same sample conserved at -80 °C).”

Materials and Methods, Statistical analysis. “The precision study was performed by calculating the coefficient of variation (CV(%) = 100 x standard deviation / arithmetic mean).”

Results: “In the precision study, the repeatability and reproducibility CVs were 2.54% and 2.98%, respectively.”

Discussion: “In relation to the precision study of the method used for the determination of HCY in pleural fluid, the repeatability and reproducibility CVs were low (<3%), so this method has a very high precision.”

3) A brief consideration of the nature of the reference standard (cytology or biopsy) is suggested. This may include a consideration of possible weaknesses in these reference standards as well as a comment on why the use of tumour markers is advantageous over these standards.

Changes in the manuscript:

Introduction: “The gold standards for the diagnosis of DPM are pleural fluid cytology and pleura biopsy. The identification of malignant cells in pleural fluid (cytology) is laborious and subjective and shows highly variable sensitivity (from 11% to 78%) [6-8]. Pleural biopsy can achieve high sensitivity, but the procedure is invasive, requires extensive experience, is technically difficult and may not be appropriate in critically ill patients [9]. In contrast, tumour markers (TMs) in pleural fluid may be measured easily and quickly in automated analyzers, being a minimally invasive technique with high accuracy for the diagnosis of DPM.”

Minor comments

1) Page 4, line 95. I suggest putting a reference for the 1993 declaration of Helsinki rather than the local and date in parentheses.

Changes in the manuscript:

Materials and Methods, Study design and patients:” This is a descriptive cross-sectional diagnostic study carried out at Puerto Real University Hospital (Cádiz, Spain) and adhered to the ethical recommendations of the Declaration of Helsinki [15].”

2) Page 6, line 151. If this population includes the members of the population assessed in reference 14 this should be stated so that the data sets are not considered to be separate for meta-analysis or review.

Changes in the manuscript:

Materials and Methods, Study design and patients:” White patients treated from January 2014 to January 2017 at Puerto Real University Hospital were studied. This population includes some members of the population assessed in our previous study [14].”

3) Page 8 table 2. There appears to be a line missing for the correlation between CA 19.9 and CA 125.

Response:

Correlation between CA 19.9 and CA 125: p>0.05 (in Table 3)

4) Page 8, data in line 172 and following. The number of significant figures seems excessive. 2 decimal places should be sufficient for the odds ratios and their CI.

Changes in the manuscript:

Results: “The odds ratios were 1.72 [95% confidence interval (CI) = 1.36–2.18] and 1.46 (95% CI = 1.14–1.87) for HCY and CEA, respectively.”

5) Page 8, line 171. Should a cutoff be supplied for the odds ratios for HCY and CEA?

Response:

In order to reduce the number of false positives, the optimal cutoff value was considered to be the one with the highest sensitivity and a specificity > 90%. The ROC curve analysis allows to choose the cut with specificity > 90%.

6) Page 9, line 199. I suspect that he AUC from this paper and the previous paper are not significantly different. I would describe them as similar rather than the previous one being slightly lower (perhaps “marginally lower” may be more accurate.

Changes in the manuscript:

Discussion: “In the recent study evaluating the pleural fluid concentration of HCY as a TM, which included 89 patients, the AUC obtained was 0.833 [14], which is similar than that obtained in this study.”

7) Page 9. Line 214. The AUC is missing a leading “0.”.

Changes in the manuscript:

Discussion: “Although CA15.3 had a high diagnostic accuracy (AUC = 0.800)…”

All authors are grateful to the editor and reviewers for their contribution in the review of this study.

---

## [Editor Report · Decision Letter 1]

4 Sep 2019

The diagnostic value of pleural fluid homocysteine in malignant pleural effusion

PONE-D-19-17055R1

Dear Dr. Santotoribio,

We are pleased to inform you that your manuscript has been judged scientifically suitable for publication and will be formally accepted for publication once it complies with all outstanding technical requirements.

With kind regards,

Luka Brcic

Academic Editor

PLOS ONE
---

## [Editor Report · Acceptance letter]

13 Sep 2019

PONE-D-19-17055R1 

The diagnostic value of pleural fluid homocysteine in malignant pleural effusion 

Dear Dr. Santotoribio:

I am pleased to inform you that your manuscript has been deemed suitable for publication in PLOS ONE. Congratulations! Your manuscript is now with our production department. 

With kind regards,

on behalf of

Dr. Luka Brcic 

Academic Editor

PLOS ONE